# Measuring the Relation between Academic Performance and Emotional Intelligence at the University Level after the COVID-19 Pandemic Using TMMS-24

Roberto Sánchez-Cabrero [1,*], Amaya Arigita-García [2], David Gil-Pareja [3], Ana Sánchez-Rico [3], Fernando Martínez-López [3] and Leonor Sierra-Macarrón [2]

1 Department of Evolutionary Psychology and Education, Universidad Autónoma de Madrid, Carretera de Colmenar Viejo, Km. 15,500, 28049 Madrid, Spain

2 Centro Universitario Don Bosco Adscrito a la Universidad Complutense de Madrid, Calle María Auxiliadora, 9, 28040 Madrid, Spain; aarigita@cesdonbosco.com (A.A.-G.); lsierra@cesdonbosco.com (L.S.-M.)

3 Department of Education, Universidad Alfonso X el Sabio, Campus de Villanueva de la Cañada, Avenida de la Universidad, 1, Villanueva de la Cañada, 28691 Madrid, Spain; dgilpar@uax.es (D.G.-P.); asancric@uax.es (A.S.-R.); fmartlpe@uax.es (F.M.-L.)

* Correspondence: roberto.sanchez@uam.es

**Abstract:** The outbreak of the global pandemic derived from COVID-19 in early 2020 has represented a huge loss of social contact for most young people. The extent of these effects is still unknown, so it is necessary to ask what the effect of this new, unforeseen, and prolonged situation on the management of emotional intelligence in university students is. This study aims to compare the academic performance, test anxiety (before and during the online exams), and emotional intelligence of 91 students in a university Master's degree program after the outbreak of the COVID-19 pandemic. The emotional intelligence was measured by the TMMS-24, the academic performance was compiled in common subjects, and test anxiety was measured by self-assessment just after finishing each online exam. The comparisons between the variables were made through means difference contrasts using Mann–Whitney U, Kruskal–Wallis, and One-way ANOVA and Spearman's rank correlation coefficient as a non-parametric test for correlational analysis. The results show that the COVID-19 pandemic has not modified how these three variables are related, so it can be concluded that the prolonged social isolation suffered by young people has not had negative repercussions on their emotional intelligence, anxiety before exams, and academic performance.

**Keywords:** emotional intelligence; test anxiety; academic performance; COVID-19; social isolation

## 1. Introduction

The outbreak of the COVID-19 global pandemic in early 2020 meant an enormous social challenge on various levels. Unpreparedness, unduly long social isolation measures, and the lack of any clear outlook regarding a short-term return to normalcy have gravely affected the psychology of millions of people [1–4].

Orgilés et al. [5] have pointed out that the prolonged absence of social contact has a significant effect on most people, particularly young people, which is at an age when social contact is imperative. A recent study by Liang et al. [6] reveals that more than 40% of youth are showing symptoms of psychological issues stemming from prolonged isolation. These results indicate that though the cause may be transitory, the symptom is not. Hence, it warrants detailed analysis and treatment.

The academic sphere is the place where this youth develops its social and emotional relations. University education, apart from a professional springboard, is also the backdrop for the socialization of students [7]. Thus, avoiding physical contact between students does not only affect them academically but also their social and psychological wellbeing.

Prolonged isolation and the youthful need to socialize compound a dangerous mix which can frequently devolve into psychological problems [6]. Academic performance is highly reliant on the social and emotional relations of the student [8]. A negative situation such as the pandemic should have negative connotations, not only socially and emotionally but also academically.

One of the most relevant factors dealing with the interrelation between mental well-being and the academic field in situations affected by isolation, which takes into account emotional and social factors and their relationship with academic performance, is emotional intelligence (EI). EI may be defined as an interrelated set of skills aimed at identifying, using, understanding, and managing our emotions, as well as those of others [9]. This concept can be mainly conceptualized from two distinct models. On the one hand, the skill model, which regards EI as a limited set of interconnected cognitive-emotional skills, is measured objectively. According to it, those skills comprise the ability to perceive, manage, facilitate and understand one's own emotions and those of others. On the other hand, the model based on traits or mixed models focused on subjective evaluations of socio-emotional skills [10].

According to the recent meta-analysis conducted by MacCann et al. [11], with more than 42,000 students coming from different educational stages, EI is the third most influential factor that affects academic performance (AP), just behind general intelligence and conscientiousness, with an approximate average influence of 20% of the variance. In the same vein, we find the recent study by Estrada et al. [12], which confirms that high levels of EI are frequently linked to better AP, and they are also commonly related to pro-social behaviors and greater citizen engagement. On the other hand, Petrides et al. [13] clarify that EI does not only have a strong positive connection with academic performance, but its benefits are usually long-lasting, and they even have positive effects on other areas, such as health, sociability, or resilience.

It is still early to understand the true dimension of the negative effects that the coronavirus pandemic leaves behind [14], but it is timely to ascertain whether this grave, unforeseen and prolonged situation can significantly affect the (EI) of university students, in particular when managing test anxiety (TA), given that final academic evaluation of learning outcomes is notoriously the moment of greatest stress and psychological vulnerability of the alumnus [15–17]. If a reduction in EI has a direct impact on the perceived TA and AP, then those who have suffered the pandemic emotionally will confront diminished academic returns as well [18].

Some scientific studies have recently emerged to shed light on this issue [14,19–21]. They all highlight the important role that EI plays in properly coping with the isolation derived from the COVID-19 pandemic. Its beneficial aspects are not just restricted to health and psychological wellbeing [20,21], but they also have a positive influence on academic stress caused by the inter-mediation of technology [16,19].

In order to efficiently cope with this situation socially and medically in a post-COVID world, it is important to identify which attributes correlate with a greater effect of prolonged isolation on EI. In their recent study with 6000+ participants (were mostly middle-aged adults between 30 and 50 years old) in Italy, Conversano et al. [22] concluded that people who were young, female, and childless were more prone to suffer more emotionally, while mindfulness is a protective factor. However, the relation between AP and the effect of prolonged isolation on EI is still unclear. Some prior studies, such as Carter et al. [23], demonstrate that there is a distinct correspondence between grades and emotional distress, which can at times even outpace intellectual capability when determining marks among the young.

Other studies, such as the one conducted by Alenezi [24], show that the relationship between EI and AP may be mediated by different attributive factors, such as gender (women respond better than their counterparts in this research), or the academic field of the studies taken. In his study, Alenezi [24] found that EI was much more relevant to science students than students with humanities backgrounds to improve their self-efficacy. On

the other hand, Abbas et al. [25], in addition to confirming the connection between EI and gender as noticed by Alenezi [24], found that age was also a factor that directly affected the perception of nostalgia and negative mood swings among university students; therefore, students were more prone to experience an anxiety crisis related to EI that affected their academic performance.

In this line, other studies confirm that the academic field of study can also affect how AP varies. Previous studies, such as that of Stoet and Geary [26] or that of O'Dea et al. [27], indicate that professional identification and gender are important factors, as they also have an effect on AP [28–31]. There is also the AP effect of the convoluted relation between STEM and gender, given that the gender-based nature of selecting a degree is evidenced both by the OECD [32] and other studies [33–35]. Nonetheless, educational equality also bears a negative effect on this gender gap [26,27].

Carrying out research on these factors implies combining the AP indicator that is a final grade with measuring TA synchronously, if possible, during the exam, given that it is a state that is completely dependent on context. Reliability, and hence validity, is drastically diminished the further from the context it is produced [16]. Finally, measuring EI can be done in multiple ways, but the 24 items version of Trait-Meta Mood Scale (TMMS-24) [36] far outstretches their adequacy because it is current, brief, and can be scalable to the young Spanish population [37].

This study has three main objectives: (1) Describe the levels of emotional intelligence (EI), test anxiety (TA), and academic performance (AP) among university students after the COVID-19 pandemic, including attributive variables based on gender, age, and academic discipline. (2) Determine whether the same attributes variables have an effect on EI, TA, and AP. (3) Validate whether there is a correlation between EI, AP, and TA among the test subjects. Successfully completing these objectives will warrant a comparison with prior research results in order to determine whether prolonged isolation due to the pandemic has indeed affected university students with characteristics similar to those of the participants in this study.

## 2. Materials and Methods

### 2.1. Participants

The study sample is composed of 91 participants voluntaries (59 women and 32 men), who were students in the year 2020–2021 of the Secondary Education Teacher Training Masters with a population of 517 students. Their mean age was 35.9 (35.08 women, 37.41 men), including a standard deviation of 7.88 (8.48 women, 6.51 men). The selection of this group for sampling is due to their heterogeneity. This Master's Program capacitates students to become Secondary Education teachers in Spain, so it includes students from an array of academic backgrounds and degrees.

The age and gender attributes of the sample are displayed in Figure 1.

The sample was drawn from the seven different academic backgrounds that enrolled in the program: Language ($n$ = 5), Mathematics ($n$ = 6), Biology and Geology ($n$ = 19), Physics and Chemistry ($n$ = 6), Technology ($n$ = 34), Geography and History ($n$ = 3) and Economics ($n$ = 18). Technology garnered the greatest subset, in contrast with Geography and History. Figure 2 charts these numbers.

Cluster sampling was carried out of a group that had in common that they were enrolled in two consecutive promotions of the *Master de Formación del Profesorado de Secundaria online de la Universidad Alfonso X el Sabio* ('Secondary Teacher Training Master's Degree' at the Alfonso X el Sabio University). No additional criteria were used to select the sample, resulting in a very variegated group regarding professional profiles, prior experience, and place of residence in Spain.

### 2.2. Instruments and Variables

In order to collect data for the study, various sources of information were used. In the three common courses of the Master's degree: "Education and social and family

environment", "Learning and personality development", and "Educational processes and contexts", for the cohort of the 2020–2021 school year, the grades obtained in the scheduled online final exam of each course. The reliability of these scores is good, based on the Cronbach Alpha measurement ($\alpha = 0.801$).

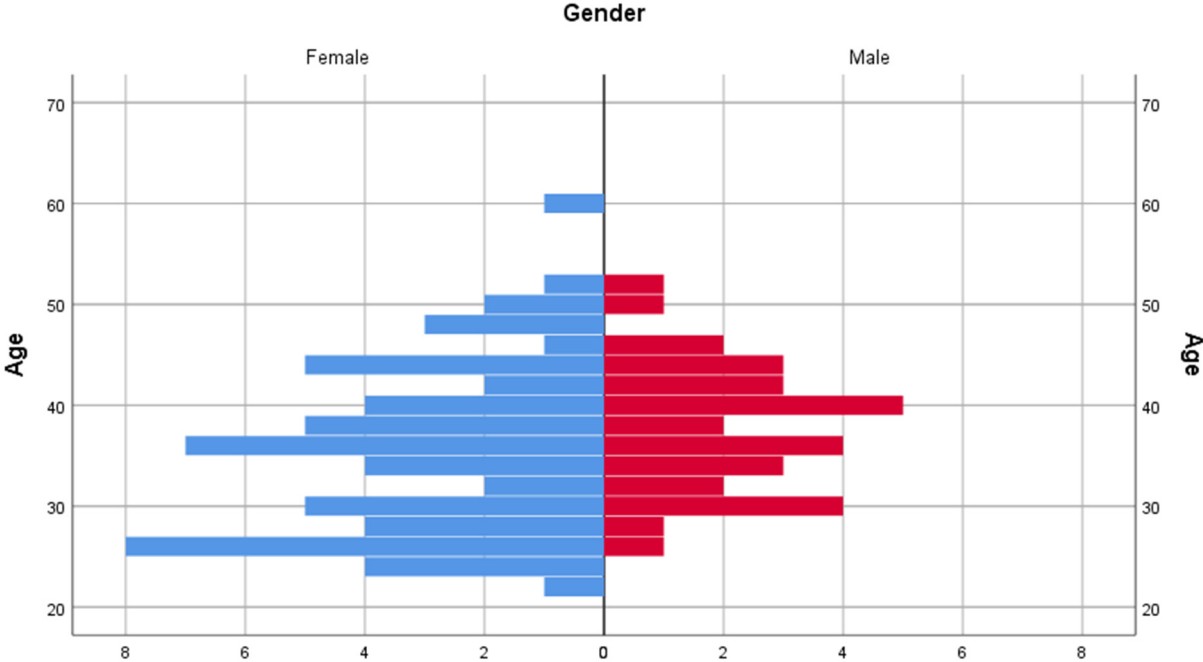

**Figure 1.** Population pyramid of the sample group.

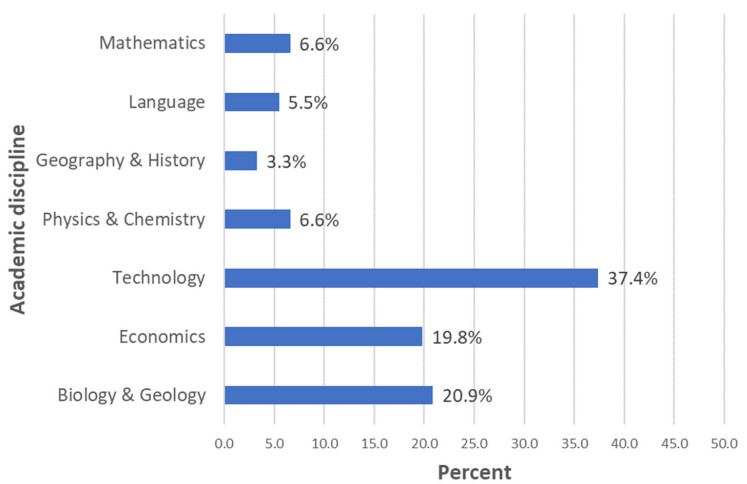

**Figure 2.** Distribution of the sample according to academic background.

A second source of information was a simple questionnaire, designed ad hoc, which gathered information on the demographic attributes of the participants (age, gender, and prior academic training), as well as the written informed consent to participate in the study.

The third and final source was the evaluation of participant EI using the 24 items version of Trait Meta-Mood Scale (TMMS-24). This is an abbreviated version devised by Fernández-Berrocal et al. [38], which utilizes the EI measuring instrument created by Salovey et al. [39] to measure perceived EI. The reduced version uses three dimensions in Spanish (attention, clarity, and repair), which are measured using 24 items in a five-point Likert scale (from 1, "totally disagree" to 5, "totally agree"). Fernández-Berrocal

and Extremera-López [40] define 'attention' as the basic ability to perceive emotions and feelings; 'clarity' is emotional understanding; 'repair' is emotional regulation [41].

According to the scale proposed by Fernández-Berrocal and Extremera-López the average and adequate scores for the 'attention' parameter are 22–32 for males and 25–35 for females, for the 'clarity' parameter are 26–35 for males and 24–34 for females, and for the 'repair' parameter are 24–35 for males and 24–34 for females. Above these scores, this scale considers it an 'excellent' score and below 'improvable'.

TMMS-24 is highly reliable both according to Cronbach's Alpha (attention, $\alpha = 0.90$; clarity, $\alpha = 0.90$; repair $\alpha = 0.86$), as well as measuring test-retest reliability: attention = 0.60, clarity = 0.70, and repair = 0.83 [42].

Finally, in order to obtain participant self-assessment on anxiety before and during an examination, two questions were included in each of the three online exams for the 2020–2021 cohort, using a single choice option in a five-point Likert scale (from 0, "no stress perceived" to 4, "totally stressed"). The questions were: 'How would you define your stress level right before taking the online exam?', to self-assess the perceived stress level at the beginning of the exam, and 'How would you define your stress level while taking the online exam?', to self-assess the level of perceived stress at the beginning of the exam. The first question was the first item of the exam, and the second question was the last item to facilitate having the self-reference fresh in the memory and consciousness of each study participant.

All the instruments used were assessed and validated by an external group: a scientific and ethics committee that verified and approved the adequacy and soundness of the experimental process (Institutional Review Board of Alfonso X the Wise University). All participants were informed and provided written consent, as recommended by the World Medical Association Declaration of Helsinki [43].

The variables used included three types: attributes, academic performance, and self-evaluation. Three attributes were used: gender (nominal variable with two categories: 'male' and 'female'), age (a discreet quantitative variable), and prior academic training (where an array of possibilities was funneled into the seven possible categories of this nominal variable: 'Biology and Geology', 'Economics', 'Technology', 'Physics and Chemistry', 'Geography and History', and 'Language' and 'Mathematics'). Academic performance (AP) variables are the result of the arithmetic mean of final online exams in the three courses mentioned above, which are common to all Master's students. This grade follows a ten-point scale. Self-assessment variables included measuring anxiety levels right before and during the exam, using a five-point Likert scale (from 0, "no stress perceived" to 4, "totally stressed"). It should be clarified at this point that the anxiety data before and during the exam were collected during the exam through two items included in the test itself.

### 2.3. Procedure and Experimental Design

The research process was carried out throughout the 2020–2021 academic year of the Master's Degree in Teacher Training, from September 2020 to July 2021. The methodological procedure followed a series of steps:

1. The attributes of the participants were extracted from their enrolment information in the Master in the 2020–2021 course in September 2020;
2. Written informed consent was obtained from the participants in February 2021;
3. Stress-perception data was collected through surveys immediately after the exams in February 2021;
4. The grades of the participants were collected in February 2021 exam session;
5. EI of participants was measured using TMMS-24 in June 2021.

The data collected permitted a correlational, comparative, sequential, and descriptive analysis of the participants. Initially, a descriptive statistical study used frequency distribution of nominal and ordinal variables, together with determining the arithmetic mean and standard deviation of the age and final grade variables.

Afterward, the Kolmogorov-Smirnov test (K-S) [44] was used to determine whether the data obtained using assessment instruments warranted further correlational and inferential parametric tests. The results can be seen in Table 1.

**Table 1.** Results of the K-S test.

| Assessment Variables | Test | Sig. |
|---|---|---|
| Academic Performance (Grades) | 0.12 | 0.004 |
| **Perceived Stress (TA)** | **Test** | **Sig.** |
| Prior to exam | 0.19 | 0.000 |
| During the exam | 0.23 | 0.000 |
| **Emotional Intelligence (EI)** | **Test** | **Sig.** |
| Attention | 0.08 | 0.200 |
| Clarity | 0.09 | 0.072 |
| Repair | 0.06 | 0.200 |

Table 1 reveals that both AP and TA do not plot out a normal distribution, indicating that parametric tests of their inferential analysis are not advisable. The three factors related to EI, however, do have normal distributions, suggesting that any inferential analysis of those results can use parametric tests.

The K-S test results encourage us to use Spearman's rank correlation coefficient as a non-parametric test for correlational analysis. Inferential analyses use Mann–Whitney U test for non-parametric testing of gender and Kruskal–Wallis for the treatment of the academic discipline variable. A parametric One-way ANOVA test will be used for EI factor analyses. Finally, Bonferroni correction was applied in order to avoid the risk of error type I when making multiple comparisons [45]. To ascertain reliability to 99% and 95% in any results obtained with the seven Academic discipline groups, we only considered a result significant when p is below $\alpha$: 0.007 for a reliability of 95% and $\alpha$: 0.007 for a reliability of 99%.

## 3. Results

The first research objective was to ascertain the levels of EI, TA, and AP among college youth, factoring in their attributes. This was done by determining both the average and standard deviation of their grades, together with the three TMMS-24 scores, as well as the anxiety self-assessment indicators. Table 2 below collects data on the arithmetic mean and standard deviation of participant age, both in total and by gender and academic discipline.

**Table 2.** Means and standard deviations of age, according to gender and academic discipline.

| Attributes | Mean | Standard Deviation |
|---|---|---|
| **Total** | 35.90 | 7.89 |
| **Gender** | **Mean** | **Standard Deviation** |
| Female | 35.08 | 8.48 |
| Male | 37.41 | 6.52 |
| **Academic Discipline** | **Mean** | **Standard Deviation** |
| Biology and Geology | 31.00 | 5.88 |
| Economics | 36.39 | 6.96 |
| Technology | 40.82 | 6.56 |
| Physics and Chemistry | 30.50 | 7.12 |
| Geography and History | 31.00 | 5.00 |
| Language | 32.00 | 10.56 |
| Mathematics | 33.17 | 8.13 |

Table 2 clearly reveals that men were much older than women. Technology majors were also older, in contrast with Physics and Chemistry, which was the youngest group. The inferential analysis was warranted in order to determine the statistical significance of these results. Table 3 emulates Table 2, but the data refers to grades instead of age.

**Table 3.** Means and standard deviations of grades, according to gender and academic discipline.

| Attributes | Mean | Standard Deviation |
|---|---|---|
| **Total** | 8.67 | 1.02 |
| **Gender** | **Mean** | **Standard Deviation** |
| Female | 8.84 | 0.77 |
| Male | 8.35 | 1.31 |
| **Academic Discipline** | **Mean** | **Standard Deviation** |
| Biology and Geology | 9.11 | 0.65 |
| Economics | 8.37 | 1.01 |
| Technology | 8.50 | 1.15 |
| Physics and Chemistry | 9.22 | 0.39 |
| Geography and History | 8.38 | 2.06 |
| Language | 8.32 | 0.72 |
| Mathematics | 9.05 | 0.72 |

Table 3 demonstrates that women get better grades than men. Regarding academic discipline, those that do STEM (Science, Technology, Engineering, and Mathematics), except for Technology, also get better grades, with an average of above 9. Once again, it is necessary to carry out inferential analysis in order to recognize the statistical meaningfulness of these results.

Table 4 will follow the same pattern as the previous two, but this time using the perceived TA, both before the exam and during an examination.

**Table 4.** Means and standard deviations of TA assessments, according to gender and academic discipline.

| | Before | | During | |
|---|---|---|---|---|
| Attributes | Mean | SD | Mean | SD |
| **Total** | 2.18 | 1.05 | 1.34 | 0.922 |
| **Gender** | **Mean** | **SD** | **Mean** | **SD** |
| Female | 2.25 | 1.03 | 1.42 | 0.855 |
| Male | 2.03 | 1.09 | 1.19 | 1.030 |
| **Academic Discipline** | **Mean** | **SD** | **Mean** | **SD** |
| Biology and Geology | 2.16 | 1.02 | 1.26 | 0.65 |
| Economics | 2.50 | 1.04 | 1.61 | 1.15 |
| Technology | 2.12 | 1.12 | 1.32 | 0.95 |
| Physics and Chemistry | 2.17 | 0.98 | 1.17 | 0.75 |
| Geography and History | 1.67 | 0.58 | 1.33 | 1.53 |
| Language | 2.00 | 0.71 | 1.40 | 0.89 |
| Mathematics | 2.00 | 1.41 | 1.00 | 0.89 |

Table 4 reveals that TA is greater for women, both before and during an examination. By academic discipline, the results were more convoluted. Economics students suffered greater TA both before and during examination. Before the exam, it was Geography and History majors that perceived less TA. However, during the examination, it was both Physics and Chemistry and Mathematics students that displayed less TA. Once again, inferential analyses were warranted before we jumped to conclusions.

Table 5 includes the results of the TMMS-24 on all three scales, both in total and by gender and academic discipline.

**Table 5.** Means and standard deviations of TMMS-24 scores, according to gender and academic discipline.

| | Attention | | Clarity | | Repair | |
|---|---|---|---|---|---|---|
| **Attributes** | **Mean** | **SD** | **Mean** | **SD** | **Mean** | **SD** |
| **Total** | 27.77 | 6.35 | 28.24 | 6.285 | 26.59 | 6.462 |
| **Gender** | **Mean** | **SD** | **Mean** | **SD** | **Mean** | **SD** |
| Female | 28.63 | 5.93 | 29.58 | 5.685 | 27.42 | 6.226 |
| Male | 26.19 | 6.89 | 25.78 | 6.676 | 25.06 | 6.705 |
| **Academic Discipline** | **Mean** | **SD** | **Mean** | **SD** | **Mean** | **SD** |
| Biology and Geology | 28.16 | 7.52 | 30.84 | 5.24 | 27.42 | 6.63 |
| Economics | 27.28 | 5.00 | 27.11 | 6.83 | 26.00 | 7.05 |
| Technology | 27.59 | 6.69 | 27.82 | 6.66 | 26.35 | 6.11 |
| Physics and Chemistry | 26.17 | 5.35 | 29.00 | 5.80 | 28.17 | 5.57 |
| Geography and History | 33.00 | 8.89 | 22.00 | 6.08 | 23.67 | 5.86 |
| Language | 28.40 | 3.05 | 26.80 | 7.12 | 24.60 | 9.07 |
| Mathematics | 27.50 | 7.29 | 29.33 | 3.56 | 28.67 | 6.83 |

Women tended to have significantly higher scores of EI, as evidenced by Table 5. By discipline, Geography and History demonstrated greater variability, with the highest Attention scores contrasting with the lowest Clarity and Repair scores. Biology and Geology students had greater Clarity, and Mathematics students scored highest in Repair. On the lower end, Physics and Chemistry students got the lowest Attention scores. This data should be analyzed inferentially in order to reach any conclusions.

The second research objective sought to find any correlations between the attributes and the EI levels, TA, or AP among the university youth in a post-pandemic world. It is important to ascertain whether differences in descriptive analyses truly correspond with statistically significant variations. The results of the inferential comparison between the various attributes are described below.

Table 6 shows the results of using non-parametric tests: Mann–Whitney U test for gender, Kruskal–Wallis for the academic discipline, and a One-way ANOVA for both, including effect size.

**Table 6.** Results of the mean comparisons of Gender and Academic discipline taking into account the quantitative variables included in the study.

| | Gender | | | Academic Discipline | | |
|---|---|---|---|---|---|---|
| **Variables** | **U** | **Sig.** | **Eta Squared** | **K-W** | **Sig.** | **Eta Squared** |
| Age | 1136.00 | 0.110 | 0.141 | 27.99 | 0.000 | 0.295 |
| Academic Performance | 782.50 | 0.179 | 0.232 | 10.29 | 0.113 | 0.108 |
| **Perceived Stress** | **U** | **Sig.** | | **K-W** | **Sig.** | |
| Before exam | 830.50 | 0.327 | 0.102 | 3.49 | 0.745 | 0.032 |
| During exam | 787.00 | 0.169 | 0.123 | 1.98 | 0.922 | 0.031 |
| **Emotional Intelligence (EI)** | **F** | **Sig.** | | **F** | **Sig.** | |
| Attention | 3.13 | 0.080 | 0.184 | 0.43 | 0.857 | 0.030 |
| Clarity | 8.17 | 0.005 | 0.290 | 1.27 | 0.281 | 0.083 |
| Repair | 2.83 | 0.096 | 0.175 | 0.41 | 0.869 | 0.029 |

U: Mann–Whitney U test; K-W: Kruskal–Wallis test.

Table 6 shows that only the EI Clarity parameter displayed significant differences by gender, proving that women manifested significant differences in this dimension. Regarding academic discipline, only age appears to find a meaningful correlation: Physics and

Chemistry students were the youngest (M = 30.5; SD = 7.12), and Technology majors were the oldest (M = 40.82; SD = 6.56).

The third and final research objective aims to establish if there was a significant correlation between EI, AP, and TA in a post-pandemic university classroom. Table 7, below, shows the results of the Spearman's rank correlation coefficient of the variables assessed, together with the age attribute (which is a discrete-continuous variable, and hence comparable).

**Table 7.** Correlations using Spearman's correlation coefficient as a contrasting statistic.

|  | Age | AP | TA1 | TA2 | EI-A | EI-C | EI-R |
|---|---|---|---|---|---|---|---|
| **Age** | - | −0.08 | 0.04 | 0.08 | −0.10 | 0.20 | 0.25 * |
| **Academic Performance (AP)** |  | - | −0.08 | −0.31 ** | 0.31 ** | 0.21 * | 0.11 |
| **TA before exam (TA1)** |  |  | - | 0.54 ** | 0.13 | −0.00 | −0.09 |
| **TA during exam (TA2)** |  |  |  | - | −0.08 | −0.13 | −0.26 * |
| **EI Attention (EI-A)** |  |  |  |  | - | 0.16 | −0.05 |
| **EI Clarity (EI-C)** |  |  |  |  |  | - | 0.58 ** |
| **EI Repair (EI-R)** |  |  |  |  |  |  | - |

\* Correlation is significant at the 0.05 level; ** Correlation is significant at the 0.01 level.

The results reflected in Table 7 indicate a strong inverse correlation between AP and TA during the exam and a direct correlation with the EI Attention and Clarity scores. In addition, EI Repair also correlated with age and EI Clarity, but inversely with TA during the exam. Finally, there was a strong direct correlation between both TA measurements, before and during an examination.

## 4. Discussion

The effect of the COVID-19 pandemic on AP has been analyzed by Sánchez-Cabrero et al. [16]. They found, when comparing the results of the 2019–2020 cohort with that of 2020–2021, that the participants from this same Master's Degree had obtained significantly better grades after the pandemic had struck (Fall 2019: M = 7.06, SD = 1.32; Fall 2020: M = 8.41, SD = 1.05). In this study, academic results were even better, with a higher average grade and less standard deviation (M = 8.67, SD = 1.02), indicating a steady progression in this sense. The reason behind this improvement is probably related to the voluntary participation of students, who are more involved in their studies. The sample does not include students who were unwilling to participate or who were less engaged with their Master's. The improvement that had been witnessed in comparison to pre-pandemic students was attributed to the way exams were designed by Sánchez-Cabrero et al. [16]. The pandemic situation warranted new and untested means of examination, which often resulted in simpler testing. This study appears to bear out that explanation as well.

The attributes that have been included in this paper appear to have little effect on AP: gender, academic discipline, or age seem to not factor in determining it. These results appear to contradict recent literature [24,25,28–31]. This may be due to the 'variability hypothesis' outlined by O'Dea et al. [27], which points out that gender-related differences may be due not to actual differences but to the over-representation of males in STEM education. The gap, therefore, is not due to a question of AP but rather to motivational aspects behind the choice of major [33–35]. By having a sample from students who are training to be teachers, an area that is predominantly female [32], this gap disappears.

TA seems to maintain similar levels as before the pandemic [46], which supports the results of other research such as Xie et al. [47] or that of Montolio and Taberner [17]. The latter found that men had lower perceived TA than women; this appears to dovetail with our results, where women have a slightly higher perceived TA, although the slight variation is statistically insignificant.

EI measurements, perhaps unexpectedly, indicate that it has not been affected by the pandemic. Using the three parameters (Attention, Clarity, and Repair) contained in the methodology outlined by Fernández-Berrocal and Extremera-Pacheco [40], we find that most indicators are average. Only Men's Clarity is close to below average, yet still within the average range (M = 25.78, close to M = 26). These results even show better EI than compared with pre-pandemic studies, such as Gaeta-González and López-García [41], which yields ostensibly worse levels. Perhaps the sample, composed of students above 35 years of age, and a more stable and mature professional profile, maybe behind this apparently high EI.

Contrast analysis of EI with other attributes only reveal significant differences in Women's Clarity; thus, we can conclude that female participants had greater emotional understanding, which partially supports the results obtained in the research by Alenezi [24] and Abbas et al. [25], who found improvements in the EI of women with respect to men. Nonetheless, Women did get higher scores on all three parameters [37], which is strongly supported by existing research [37–39,41].

In the final part of the analysis, an attempt to find correlations between age, AP, TA, and EI closely echoes previous studies such as that of Thomas et al. [48]. We have found a significant inverse correlation between AP and TA during the exam and a direct correlation with EI Attention and Clarity, which demonstrates the logical conclusion that a bad grade in the exam is due to the negative emotions experienced during the exam, such as anxiety. On the other hand, the EI parameters Repair and Clarity are directly correlated, which was also found in other studies [37,49]. Age, however, only correlates with EI Repair, clearly a demonstration that age facilitates emotional maturity [25,50,51].

Finally, there is a strong direct correlation between TA before and during examination in the same subject, which is an expected result since there tends to be a coherence in the perceived TA included in the surveys.

## 5. Conclusions

The outbreak of the SARS-Coronavirus-2 pandemic and the ensuing quarantine brought upon everyone a highly unexpected and stressful time, with a strong deterioration of the emotional condition of many people. This study demonstrates, however, that the relation between EI and AP was not altered with respect to pre-pandemic times, and the relation with the usual social factors remained unchanged.

The various attributes considered in this study, and their relationship with AP, TA, and EI, saw no variation with respect to previous studies. The psycho-educational consequences of the pandemic are the same regardless of age, gender, or academic discipline. The coronavirus changed education enormously, but it has not affected the weight of other attributes, at least in Spain and with the demographic attributes of this sample.

## 6. Limitations

This study was conducted under particular circumstances that might limit the generality of the conclusions drawn here. Firstly, we should consider that in order to corroborate these results, a larger sample would be necessary, including extending it to other student groups. The final sample of the present investigation, consisting of 91 participants, represented 17.6% of the total number of students in the year 2020–2021 of the Secondary Education Teacher Training Masters. However, to broaden the scope of the conclusions to other groups, a significantly larger sample would be required.

Secondly, we should note that the sample at hand had an average of 35+ years, which implies a degree of maturity and experience above the average of university students in most investigations. It is possible that this maturity makes them be more aware of their cognitions and emotions, and therefore, their perception of the interconnected aspects such as EI and TA may differ from younger population groups.

Finally, we should clarify that this is a highly heterogeneous sample; thus, the subgroupings according to attributes, especially academic discipline, result in very small

numbers. Hence, we cannot take the results of this study lightly and generalize them to other academic groups in which the sample is very scarce.

**Author Contributions:** Conceptualization, R.S.-C., A.A.-G. and D.G.-P.; methodology, R.S.-C., A.A.-G. and A.S.-R.; validation, R.S.-C., A.A.-G., D.G.-P., A.S.-R., F.M.-L. and L.S.-M.; formal analysis, R.S.-C., D.G.-P. and A.S.-R.; investigation, R.S.-C., D.G.-P. and A.S.-R.; resources, R.S.-C., A.A.-G., D.G.-P., F.M.-L. and L.S.-M.; data curation, R.S.-C., A.A.-G. and F.M.-L.; writing—original draft preparation R.S.-C., A.A.-G., D.G.-P., A.S.-R., F.M.-L. and L.S.-M.; writing—review and editing, R.S.-C., A.A.-G., D.G.-P., A.S.-R., F.M.-L. and L.S.-M.; visualization, R.S.-C., A.A.-G. and D.G.-P.; supervision, R.S.-C. All authors have read and agreed to the published version of the manuscript.

**Funding:** This research received no external funding.

**Institutional Review Board Statement:** The study was conducted in accordance with the Declaration of Helsinki and approved by the Institutional Review Board of Alfonso X the Wise University (protocol code 02, 1 March 2021) for studies involving humans.

**Informed Consent Statement:** Informed consent was obtained from all subjects involved in the study.

**Data Availability Statement:** The data presented in this study are available on request from the corresponding author.

**Conflicts of Interest:** The authors declare no conflict of interest.

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
