# Peer review of "Measuring the Relation between Academic Performance and Emotional Intelligence at the University Level after the COVID-19 Pandemic Using TMMS-24"

_sustainability, doi:10.3390/su14063142_

Round 1

Reviewer 1 Report

This paper considers interesting and relevant issues regarding the effect of the pandemic and the consequent loss of social contact on mature age students. The authors have chosen to consider test anxiety, academic performance, and emotional intelligence. It was reassuring in general to read that the results of this study indicate no bad impacts on this group of participants in contrast to expectations and some other literature.

I noticed some minor English issues and typos; for instance, line 241 and line 293.  At lines 52 to 56, I was surprised to see a short paragraph in another language which I don’t think was supposed to be there.

Overall, I wondered why emotional intelligence has been chosen here and it would be good to see greater clarity in the introduction about why emotional intelligence might be thought to be impacted by social isolation. I could imagine why isolation might make people depressed or have other psychological impacts, but the link with EI (which is after all about awareness of own and others’ emotions and regulation of others and own emotions) didn’t seem intuitive to me. I can more readily understand why test anxiety and academic performance might have been affected by the isolation. Overall, I think the introduction is a bit brief, I think there needs to be more information in more detail about other authors’ findings and the coverage didn’t seem very logical to me. Also please spell out TMMS-24 when it first appears.

At line 130, what is the nature of the simple questionnaire? I think you are talking about demographic information. Regarding test anxiety, what were the questions asked? In line 143 you say that test anxiety was assessed before and after the examinations, but in line 169 you say it was assessed immediately afterwards, and then in table 4 we are again reading about before and during. I would like to see some clarification here. The title of table 6 is too long.

At line 283 there is a mention about mean scores for the EI measurement: I would have liked to have read earlier about what is the average range for the 3 dimensions of EI used in your study.

You’re talking about “after the pandemic” and “post pandemic” but we aren’t actually over the pandemic yet. There have been several waves of course and I think it would be helpful to know the dates when this research was conducted: I didn’t notice a mention about this.

Author Response

Thank you for all the contributions made, which have helped to greatly improve the manuscript. All the reviewers' requests have been met in an extremely short time, despite the complexity of the work requested.

The main improvements are described below:

  1. All typos and bugs flagged by reviewers have been fixed.
  2. A revision of the translation has been carried out.
  3. The Introduction section has been improved, expanded, and properly justified, with numerous new citations, definitions of the concepts involved, and greater clarity of ideas.
  4. The Method section has been detailed, clearly specifying all the poorly described and improvable parts.
  5. Effect size in the contrasts of means has been calculated in the Results section.
  6. Discussion and Conclusions have been slightly improved and expanded.
  7. A new Limitations section has been created, specifying what the possibilities are to be able to generalize the results and the main limitations of the results obtained.
  8. Numerous new references have been added to justify and frame the research.
  9. Other minor changes requested by reviewers.

In total, 1400 words (25% more) and 13 new references have been added, leaving a much more robust and complete project thanks to the excellent contribution of the reviewers. In the attached document you can see the main improvements made in red.

Reviewer 2 Report

I carefully read the paper entitled "Measuring the relation between academic performance and emotional intelligence at the university level after the CoViD-19 pandemic using TMMS-24" and I found it too weak to recommend publication. The main two issues that made me decide in this way are: the lack of justification both theoretically and empirically for this work and a lack of statistical power to have robust results. 

First, the paper completely lacks a theoretical motivation for why emotional intelligence should be impacted by the pandemic. Moreover, no distinction has been made regarding the different conceptualizations of emotional intelligence (e.g., ability-based models, trait EI models). 

The authors presented some empirical evidence that however fails to justify in a coherent way why the authors are testing their hypotheses. 

The sample size is not justified and 91 participants are not enough to achieve enough statistical power. 

"Inferential analysis is warranted in order to determine the statistic significance of these results." (Line 199). So, please, do them. Also for other results like: "Table 3 demonstrates that women get better grades than men". A statistical comparison should be made and a proper effect size computed, especially for results in Table 5. In any case, 5 tables for just descriptive statistics are redundant, to say the least. 

Lines 52-56 are not in English and are probably a repetition of subsequent lines. 

Author Response

(The authors gave the same response as above.)

Reviewer 3 Report

The authors’ work addresses issues of emotional intelligence (EI), test anxiety (TA) and academic performance (AP) under the effect of COVID-19 pandemic. The significance of the work also corresponds to the theme of this important focused Special Issue, and expected to add to the evolving collection of studies investigating relevant impact of the pandemic. With the learning modalities and social contact affected significantly by this prolonged and unprecedented pandemic period, it is worth investigating the impact on emotional intelligence and academic performance of the youth. The study adopts a quantitative approach adopting TMMS for EI measurement, and there are substantial efforts observed for the statistical analysis and multiple comparison. The need of the current study is also supported by cited examples of recent investigations and the ongoing development of pandemic. While the significance of the work is appreciated, there are some important issues needed to be addressed. Please refer to the following details.

1.

It is described on page 6, ln 197: “Table 2 clearly reveals that men are much older than women.” Looking at the details of the corresponding table, it was observed that only the opposite can be deduced from the table, i.e. women participated in the study are older than men (as reflected by the figures showing that female has a mean age of 37.41 and male has an average of 35.08). The information reflected from the table is also not consistent with the corresponding description on page 3 ln 104.

Similarly, on page 6, ln 202, the description of “women get better grades” contradicts to what is being reflected from Table 3 (with male having a mean score of 8.84 and female with a mean score of 8.35). For the TA assessment, from Table 4 on page 7, male has a mean of 2.25 while female gets a lower mean of 2.03 (assessed before examination). This also contradicts to what is being described on page 7, ln 210 (…reveals that TA is greater for women…).

And for Table 5 on page 7, the scores of all EI items are lower compared to those of male, which are contradicting with the corresponding explanation on page 7, ln 220, “Women tend to have significantly higher scores of EI”, as well as other relevant explanations that follow.

2.

The work addresses significant issues of AP and EI in the pandemic period. Drawing insights from the experience in the current study is important for comparing results with other studies and for future research basis. I acknowledge the authors for mentioning future work based on some limitations. But they are just briefly described in the last part of concluding paragraph of the manuscript. I suggest having a formal section of limitation for more details on the limitations, to be written at the end of discussion part, considering more details such as the sample size, age and student group, variegated profile, and prior experience of participants. The formal mentioning of limitations would further help for reference when taking into account of comparisons with related studies, and also helpful for drawing insights based on the experience of this study.

3.

Appreciate the team for extensive efforts in the statistical manipulations and considering inferential analysis and parametric tests. There are efforts made for correlational analysis in multiple comparisons. May consider supplement more details on the tests applied. The use of Bonferroni correction can be supported with explicit statement regarding the need and more descriptions or examples on the specific hypotheses tested and corrections made for adjustment of p values (page 5, ln 187-189). Or the authors may consider adding relevant references for the tests applied, which would be helpful considering the wide readership of the Journal. For example, page 5, ln 175, for the Kolmogorov-Smirnov test, may add relevant reference:

Massey Jr, F. J. (1951). The Kolmogorov-Smirnov test for goodness of fit. Journal of the American statistical Association46(253), 68-78.

4.

Page 3, ln 98-99 / page 2, ln 70-78, regarding the related studies or prior research results, if the details, such as the academic level and participant group details of previous studies, can be mentioned explicitly, for readers’ better reference on the similarity and difference of reported studies in comparison to the current work. For example, the current work has participant group of mean age 35, enrolling in Master’s programme. Can mention more the major focus of other studies or supplementing further references, in a variety of student group nature, to make more clear how the current work also contributes to the collection of studies in this evolving pandemic situation. This is expected to help for understanding more the background and significance of the current study for drawing relevant insights from described results.

In addition, there are further minor suggestions, including some examples of refinement which could be done in a formal prior proofreading. Please refer to the following comments.

1.

Consistency of specific terms should be addressed. COVID-19 was used in abstract and parts of the manuscript. Would consider a consistent use for other parts written as CoViD-19, e.g. in the title; page 1, ln 37; page 2, ln 70; page 2, ln 94; page 8, ln 253.

Also, in writing the objectives (page 2, ln 94-95), attributes variables in ln 95 would be revised the same as attributive variables in ln 94.

2.

Accuracy of writing needs to be checked.

The part on page 2, ln 52-56, which is written in Spanish, should be deleted, as I suppose the lines that follow (ln 57-61) are the translation in English.

Page 9, ln 281: “…outlined by Fernandez-Berrocal & Extremera-Lopez [29]…”, should the latter author be Extremera-Pacheco instead?

3.

Some other minor suggestions for enhancing presentation and writing:

Although it can be inter-converted between the number and the percentage of each academic discipline, would consider adding the absolute number of enrollment in Figure 2 (page 4) or a separate table showing the respective number and percentage of each category.  

Page 6, ln 203, for the mentioning of STEM, can add here a bracket reference to include the four academic disciplines identified as STEM subjects in this study).

Page 8, Table 6, in the caption can put (U) after Mann-Whitney U test, and (K-W) after Kruskal-Wallis as a key for the symbols displayed in the table.

Page 8, ln 241, “tehre” should be “there”.

Page 8, ln 254, comma after [10] should be a full-stop.

Page 9, ln 267, there should be a comma between the two categories, which is written as “gender academic discipline”.

Page 9, ln 278: missing of full-stop after the end of the paragraph.

Page 9, ln 293: “studies” to replace “sties”.

Page 2, ln 94, “based on” to replace “base on”.

Page 5, ln 157, should leave a space before “Academic”.

Page 8, Table 6, the short form of Emotional Intelligence should be EI instead of IE.

Author Response

(The authors gave the same response as above.)

Round 2

Reviewer 1 Report

The authors are done a good job in addressing suggested changes and including considerable relevant literature justification upfront. I just noticed something strange at line 342, very easily fixed. Well done on the modifications

Author Response

Thank you for your kind words and for the effort you put into reviewing our project. If our scientific project is finally publishable and valuable for the scientific community, it is largely due to your involvement ;)

Reviewer 2 Report

The authors answered my previous point about justification (even not totally), however, they completely ignored the issue about sample size. 91 participants are simply too few to conduct the statistical analyses that they did. Thus, the study is severely underpowered in my opinion and I cannot recommend the publication. 

Author Response

I respect and accept your criticism. Thank you for your effort in evaluating our project. It has been a huge learning for us.

Reviewer 3 Report

The authors have well addressed my comments and suggestions. I greatly acknowledge the substantial work in the enhanced explanations and references, as well as the added limitation section. There were good efforts to enhance the manuscript. I only have a few remarks for reference in possibility of proofread in the future.

  1. The title: COVID-19;
  2. Page 10, ln 350: “sites” should be replaced by “studies”;
  3. Page 6, ln 239: the name should be “Bonferroni”, not “Bonferrroni”.

I thank the authors again for the extensive work done to address my review.

Author Response

(The authors gave the same response as above.)
